# NIR-II AIEgens with Photodynamic Effect for Advanced Theranostics

**DOI:** 10.3390/molecules27196649

**Published:** 2022-10-06

**Authors:** Shuai Yin, Jianwen Song, Dongfang Liu, Kaikai Wang, Ji Qi

**Affiliations:** 1School of Pharmacy, Nantong University, Nantong 226001, China; 2State Key Laboratory of Medicinal Chemical Biology, Frontiers Science Center for Cell Responses, Key Laboratory of Bioactive Materials, Ministry of Education and College of Life Sciences, Nankai University, Tianjin 300071, China; 3School of Chemistry and Life Science, Changchun University of Technology, Changchun 130012, China

**Keywords:** aggregation-induced emission, NIR-II, photodynamic, phototheranostic, precision medicine, biomedical application

## Abstract

Phototheranostics that concurrently integrates accurate diagnosis (e.g., fluorescence and photoacoustic (PA) imaging) and in situ therapy (e.g., photodynamic therapy (PDT) and photothermal therapy (PTT)) into one platform represents an attractive approach for accelerating personalized and precision medicine. The second near-infrared window (NIR-II, 1000–1700 nm) has attracted considerable attention from both the scientific community and clinical doctors for improved penetration depth and excellent spatial resolution. NIR-II agents with a PDT property as well as other functions are recently emerging as a powerful tool for boosting the phototheranostic outcome. In this minireview, we summarize the recent advances of photodynamic NIR-II aggregation-induced emission luminogens (AIEgens) for biomedical applications. The molecular design strategies for tuning the electronic bandgaps and photophysical energy transformation processes are discussed. We also highlight the biomedical applications, such as image-guided therapy of both subcutaneous and orthotopic tumors, and multifunctional theranostics in combination with other treatment methods, including chemotherapy and immunotherapy; and the precise treatment of both tumor and bacterial infection. This review aims to provide guidance for PDT agents with long-wavelength emissions to improve the imaging precision and treatment efficacy. We hope it will provide a comprehensive understanding about the chemical structure–photophysical property–biomedical application relationship of NIR-II luminogens.

## 1. Introduction

Theranostics that concurrently integrates accurate diagnosis and in situ therapy into one platform represents an attractive approach for accelerating personalized and precision medicine [1,2,3,4]. Light represents a promising modality for disease diagnosis and treatment for the salient merits of excellent spatiotemporal resolution, real-time control, noninvasive feature, and portable instruments [5,6,7]. Recently, phototheranostics has attracted considerable attention from both the scientific community and clinical doctors [8,9,10]. Fluorescence imaging possesses very high sensitivity, and photoacoustic (PA) imaging is a new technique with excellent penetration depth and spatial resolution [11,12,13,14,15]. Phototherapy methods, such as photodynamic therapy (PDT) and photothermal therapy (PTT), have shown great promise for treating many diseases [16,17,18,19]. During these treatments, the Jablonski diagram determines the photophysical energy transition processes that closely associate with the various phototheranostic properties of an agent [20,21,22,23]. For example, fluorescence emission originates from the radiative decay in the singlet excited state; photothermal effect and the PA signal are related to the nonradiative decay pathway; and PDT describes the reactive oxygen species (ROS) generation in the triplet excited state [24,25,26,27]. Therefore, the utilization and manipulation of these energy transformation processes are of vital importance for boosting the disease treatment efficacy. A lot of materials have been explored for the light-related theranostics, which include carbon nanomaterials, metal nanostructures, rare earth metal-doped systems, quantum dots, and organic molecules [28,29,30,31,32,33]. Among them, organic materials hold some intrinsic advantages of good biocompatibility, well-defined structure, facile chemical modification, and easily tunable properties [34,35]. For example, the organic near-infrared (NIR) dyes indocyanine green (ICG) and methylene blue (MB) have been approved by the FDA for clinical use; and NIR fluorescence-guided surgery has proved to be a highly efficient method for improving the accuracy of cancer resection [36,37]. However, most conventional fluorescent dyes face the aggregation-caused quenching (ACQ) problem, which considerably declines the optical properties, such as fluorescence emission and PDT; thus, this limits their real applications [38,39].

In 2001, Tang and coworkers first coined a new photophysical phenomenon, aggregation-induced emission (AIE) [40], which represents a kind of emitters that show no or weak emission in dilute solution; however, the emission greatly intensifies in aggregate form (Figure 1a). The underlying working mechanism of AIE luminogens (AIEgens) is that the propeller-like molecular structure could consume the excited state energy via intensive molecular motion in solution through a non-radiative decay process, while the molecular motion is significantly restricted in aggregate state; thus, the radiative pathway is open [41,42,43]. Based on this guideline, a library of AIEgens with emission colors ranging from UV–Vis to the NIR spectral region have been developed; and they have found applications in many fields, such as optoelectronic devices, chemosensing, microstructure visualization, and biomedical applications [44,45,46,47,48,49]. Not only is the fluorescence of AIEgens intensified in aggregate, but some related imaging/therapeutic properties would also be altered. For example, the light-triggered ROS generation ability will be boosted in the aggregate state, as compared with that in solution [50,51,52]. The violent molecular motion could also help to promote the photothermal and PA transition, benefitting PTT and PA imaging [53,54,55,56]. Accordingly, AIEgens hold great promise for biomedical applications.

Another obstacle for light-related diagnosis and therapeutics is the limited penetration depth and spatial resolution in vivo. The light-tissue interaction (e.g., autofluorescence, absorption, scattering, and reflection) is the determinant factor for optical imaging, which is highly affected by the wavelength [57,58]. Both the penetration ability and imaging quality increase as the light wavelength becomes longer. For instance, the conventional transparent NIR window (NIR-I, 700-900 nm) could realize much better in vivo biological imaging than the UV and visible spectral region [59,60]. In the recently emerging second NIR (NIR-II, 1000–1700 nm) biological window, the light-tissue interaction is greatly reduced; thus, very high penetration and resolution could be realized (Figure 1b) [61,62,63,64]. NIR-II imaging and therapy have shown promising applications in many areas, such as vascular diseases, tumors, and brain diseases [65,66,67,68,69]. During the last several years, some NIR-II AIEgens have been developed, which opens up a new avenue for obtaining an ultrabright NIR-II nanoagent [70,71,72,73,74,75]. In addition to the excellent imaging of NIR-II AIEgens in the living body, it is highly desirable to endow them with a therapeutic property (e.g., PDT) to enable precision medicine. Nevertheless, it is usually difficult to confer a PDT property on NIR-II fluorophores as the energy levels are relatively low [76,77]. Recently, several NIR-II AIEgens with a PDT function have been developed for biomedical applications. Moreover, NIR-II emitters possess long-wavelength absorption and a small-energy bandgap, which is naturally born with nonradiative thermal deactivation [78,79,80,81]. Thus, other properties including a PTT and PA signal would be easily obtained from NIR-II emitters, enabling multifunctional phototheranostics.

**Figure 1 molecules-27-06649-f001:**
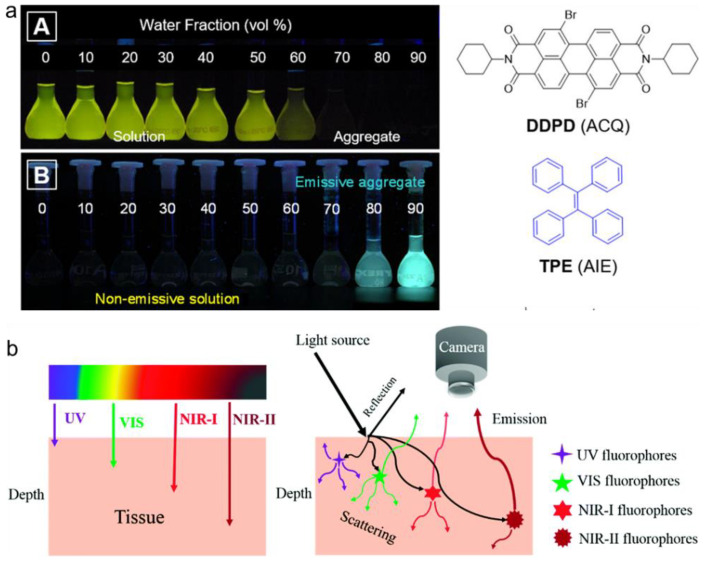
(**a**) Chemical structures of typical ACQ and AIE molecules; and photographs of them in solution and different aggregate states under UV light. DDPD: *N*,*N*-dicyclohexyl-1,7-dibromo-3,4,9,10-perylenetetracarboxylic diimide, TPE: tetraphenylethene (Reprinted with permission from Ref. [43]. Copyright 2018, American Chemical Society). (**b**) Schematic illustration of the penetration depth of different spectral regions and light-tissue interaction (Reprinted with permission from Ref. [63]. Copyright 2018, The Royal Society of Chemistry).

In this review, we summarize the recent advances of NIR-II AIEgens with PDT as well as other related optical properties for biomedical applications, mainly focusing on the photodynamic AIEgens with a maximal fluorescence emission wavelength above 900 nm (Figure 1). We will discuss the molecular design strategy for manipulating the response spectral region and photophysical energy transition process. Most of the photodynamic NIR-II AIEgens possess multifunctional properties (e.g., fluorescence, PA, PDT, and PTT), which provide a good opportunity for precise diagnosis and therapy. The phototheranostic applications in living bodies are presented, which are mainly about tumor imaging and treatment. Finally, the future challenges and perspectives of photodynamic NIR-II AIEgens are also discussed. This review aims to provide guidance for the PDT agents with long-wavelength emission to improve the diagnostic precision and treatment outcome. It will provide a comprehensive understanding about the chemical structure–photophysical property–biomedical application relationship of NIR-II luminogens, especially the multifunctional systems for precision medicine.

## 2. Photodynamic NIR-II AIEgens for Subcutaneous Tumor Phototheranostics

There are mainly two types of photodynamic mechanisms [82,83,84,85]. For type-I PDT, electron and/or proton transfer occurs between the triplet excited state and adjacent substrates; generating free radicals that could further react with other molecules (e.g., water and oxygen) to produce ROS, such as hydrogen peroxide (H_2_O_2_), superoxide anion radical (O_2_^•−^), and hydroxyl radical (HO^•^). For type-II PDT, the energy of triplet excitons transfers to ^3^O_2_ to form singlet oxygen (^1^O_2_), in which the energy of the lowest triplet excited state (T_1_) should be higher than that of oxygen sensitization (0.98 eV) [86,87,88]. The type-I mechanism is less oxygen-dependent than type-II PDT, which is therefore considered to be suitable for applications in a hypoxic environment. This explains why the NIR-II emitters with a PDT property are rare and precise molecular design should be conducted. Most NIR-II fluorophores have relatively low bandgaps of < 1.5eV, in which the efficiency of radiative decay is low and the nonradiative decay becomes dominant [89,90,91]. Thus, the nonradiative thermal deactivation-associated photothermal and PA effect are usually observed for NIR-II agents, which enables multifunctional properties.

In 2020, Xu et al., reported the positively charged photothermal photosensitizers with a donor–acceptor (D–A) structure that could emit NIR-II light for fluorescence/PA/photothermal tri-modal imaging and PDT/PTT therapy of a tumor [92]. They synthesized and compared a series of D–A molecules with different conjugated thiophene-based spacers (Figure 2). Thiophene, as an electron-rich heterocycle, could not only increase the electron-donating property, but also extend the conjugation length. As a result, the compound bearing the two thiophene units exhibited bathochromic shift in the fluorescence spectrum with a peak at about 950 nm. Studies on the ROS generation capability and photothermal efficiency of the three AIEgens revealed that the ROS generation rate was TSSAM > TSAM > TAM. In addition, among the three AIEgens, TSSAM also had the best photothermal effect with a photothermal conversion efficiency (PCE) of 40.1%. These results indicated that TSSAM was an excellent agent with NIR-II emission, high ROS generation, and high PCE. TSSAM NPs were evaluated for multimodal imaging and therapeutic performance in 4T1 tumor-bearing mice. The NIR-II fluorescence and PA signals remained strong with good spatio-temporal resolution at 24 h post injection. The following synergistic PDT and PTT treatments under imaging guidance could successfully eliminate subcutaneous tumors and inhibit tumor metastasis with high tumor-killing efficiency by only one injection and one-time irradiation.

Almost at the same time, Tang and Wang et al., reported a powerful AIEgen with all the phototheranostic modalities, including NIR-II fluorescence, PA and photothermal imaging, PDT and PTT treatments (Figure 3) [93]. They designed three compounds (TI, TSI, and TSSI) consisting of 1,3-bis(dicyanomethylene) indole as the A unit, TPA as the D unit, and a thiophene segment as both π-bridge and electron donor. The thiophene spacer in TSI and TSSI resulted in a significant red shift of the absorption/emission wavelength; and the gradually increased D–A intensity in TI, TSI, and TSSI also significantly enhanced the ROS production and photothermal conversion. As a result, TSSI NPs exhibited the reddest photoluminescence (PL) wavelength, best PDT property, and highest PCE (46%). The encapsulated TSSI NPs could be internalized into lysosomes in 4T1 cancer cells, which exhibited excellent anti-tumor properties under 660 nm laser irradiation. Inspired by the multifunctional phototheranostic properties, the in vivo biomedical applications of TSSI NPs were carried out in 4T1 tumor-bearing BALB/c nude mice. The NIR-II fluorescence imaging helped to visualize the tumor site sensitively, which was also confirmed by the PA signal and photothermal imaging. Subsequently, the in vivo tumor killing activity of TSSI NPs was verified, and the tumor was effectively eradicated with only one injection and one-time light irradiation without recurrence. In comparison with the traditional “all-in-one” strategy, this “one-for-all” agent based on an AIEgen achieves multifunctional phototheranostics in the more direct manner and maximizes the efficacy of light therapy.

Zhang et al., synthesized a series of AIEgens with distorted tetraphenylethylene (TPE) and diphenylamine (DPA) as donors and molecular rotors, electron-rich carbazole as a conjugated bridge, and three different positively charged fractions (pyridine, quinoline, and acridine) as acceptors, affording three derivatives: TPEDCPy, TPEDCQu, and TPEDCAc [94]. By increasing the electron-withdrawing capacity of the A units from pyridine to acridine, the intramolecular charge-transfer effect was significantly enhanced and the intramolecular motion was more intense, which led to red-shifted absorption in the NIR-I region and strong emission in the NIR-II region (Figure 4). TPEDCPy, TPEDCQu, and TPEDCAc showed absorptions in the range of 460–580 nm, corresponding to maximum emissions of 660 nm, 730 nm, and 980 nm with a fluorescence quantum yield (QY) of 2.9%, 2.6%, and 0.4%, respectively. The ROS generation capacity was also evaluated by using 2’,7’-dichlorofluorescein diacetate (DCFH-DA) as the indicator, which decreased in the trend of TPEDCPy < TPEDCQu < TPEDCAc. Further verification of the photothermal behavior of the three molecules showed that TPEDCPy and TPEDCQu almost had no temperature increase, while TPEDCAc displayed good photothermal effect with a PCE of 44.1% as the large acridine part provided more space for backbone deformation and rotor torsion. Noteworthy, the amphipathic TPEDCAc could easily self-assemble into homogeneously distributed nanoaggregates with good stability. Based on the bright fluorescence emission in the NIR-II region of TPEDCAc aggregates, the high photodynamic/photothermal efficiency, and the excellent tumor-killing effect at the cellular level, the tumor diagnostic and therapeutic efficacy in vivo was evaluated. The fluorescence signals in the NIR-II region started to appear in the tumor site at 0.5 h post injection, which became strongest at about 6 h and the tumor could still be clearly visualized at 48 h. Moreover, the concurrent PA imaging in vivo helped to provide more detailed information about the tumor. Under 660 nm laser irradiation, the temperature of the tumor site increased significantly, and the combination of PTT and PDT treatment successfully suppressed the tumor metastasis in MCF-7 tumor-bearing mice.

## 3. Photodynamic NIR-II AIEgens Combined with Other Therapeutic Modality

The cisplatin-based neoadjuvant chemotherapy (NAC), which plays an important role in combined surgical resection against microscopic and diffuse cancer cells, has become a standard of care and is supported by the latest clinical practice guidelines [95,96]. However, the dose-dependent toxicities of cisplatin-based NACs significantly hinder their applications. To solve this problem, Ding et al., developed a light-enhanced cancer chemotherapy (PECC) strategy based on an AIEgen. The biocompatible and biodegradable bovine serum albumin (BSA) was used as a nanocarrier to load AIEgen (BITT) and the cisplatin (IV) prodrug Pt-2COOH (DSP) to construct a NIR-II fluorescence-guided PECC-based drug for the treatment of bladder cancer (Figure 5) [97]. BITT exhibited maximal absorption/emission at 594/906 nm, and the molecule had good fluorescence and photothermal properties with a fluorescence QY of 2.42% and PCE of 36.7%. By simple mixing in an aqueous solution, BITT induced self-assembly of BSA after coupling with platinum (IV) to form stable BITT@BSA-DSP NPs without additional cross-linking agents. The prepared BITT group aggregates were entangled in the hydrophobic microenvironment of the BSA nanocages and this rigid backbone structure reduced the torsional rotation of BITT, which facilitated the radiative pathway due to the RIM mechanism and inhibition of the non-radiative thermal inactivation pathway (PCE = 26.4%), enhanced fluorescence emission (QY = 4.64%) and ISC to produce toxic ROS. In vitro and in vivo experiments validated that the integrated NIR-II fluorescence imaging-guided PECC could effectively promote bladder cancer sensitivity to cisplatin chemotherapy, significantly inhibiting bladder cancer progression by reducing tumor cell proliferation and promoting apoptosis.

Jiang et al., developed an AIEgen-based multifunctional therapeutic nanoplatform that integrated NIR-II fluorescence, photodynamic, photothermal, and immune effect [98]. The DDTB molecule possessed a highly distorted structure and branched conformation, which could effectively inhibit the intermolecular π-π stacking, and enable a remarkable AIE characteristic (Figure 6). The DDTB-DP NPs had a distinct absorption peak at 687 nm and an emission peak at 973 nm with a Stokes shift as large as 286 nm. The fluorescence QY of the NPs was calculated to be 0.96%, and DDTB-DP NPs had a good photothermal effect with a PCE of 30.7%. Interestingly, ROS was also efficiently produced under 660 nm laser irradiation, and DDTB-DP NPs had a higher ROS generation capacity than DDTB, with a ^1^O_2_ production quantum yield of 1.05%. After injecting into HeLa tumor-bearing mice, a strong NIR-II fluorescence signal from DDTB-DP NPs was observed at the tumor site. Then, DDTB-DP NPs-mediated PTT/PDT was conducted on the residual small tumors after surgical resection; which indicated that the combination of traditional surgery and light-dependent PTT/PDT could overcome their limitations and had great clinical application potential. Finally, the NPs + PD-L1 antibodies under light were evaluated to study whether they could improve the anti-tumor immune effect of primary tumors in mice. The results showed that most tumors could be ablated without recurrence with the treatment of “NPs + light + PD-L1 antibody”. These results revealed that the combination of DDTB-DP NPs-mediated PTT/PDT and PD-L1 antibodies could achieve satisfactory tumor immunotherapy performance.

## 4. Photodynamic NIR-II AIEgens for Orthotopic Tumor Phototheranostics

For the shallow penetration depth of most light, the applications are mainly limited to subcutaneous tumors [99,100,101,102]. The improved penetration ability of NIR-II light makes the phototheranostics of orthotopic tumors possible. Chen et al., reported a NIR-II AIEgen for image-guided tumor resection and phototherapy of orthotopic liver cancer [103]. They synthesized a D-A-D-structured AIEgen, 7,7ʹ-(6,7-diphenyl-[1,2,5]thiadiazolo [3,4-*g*] quinoxaline-4,9-diyl)bis(10-octyl-10*H*-phenothiazine) (PTZ-TQ), which had maximal absorption at 650 nm, and a significant NIR-II fluorescence emission peak at 1150 nm with a fluorescence QY of 0.3% (Figure 7). PTZ-TQ NPs also exhibited a high ^1^O_2_ production quantum yield of 10% using ICG as the reference (12%). The NIR-II imaging and PDT capacities were evaluated in a nude mouse orthotopic liver tumor model. A remarkable NIR-II fluorescence signal at the liver site was recorded, which was capable of identifying the boundary between the tumor and normal liver organ. In a clinic, sometimes tumor resection cannot be performed due to the presence of many smaller tumors or a very large tumor, and multiple treatments are the common treatment methods in clinical practice. Given the strong ROS generation capacity of PTZ-TQ NPs, it was used for PDT of the residual tumor after surgery, achieving complete suppression of orthotopic tumors without recurrence.

Recently, Li et al., constructed a photosensitizer DCTBT with an AIE signature through increasing the D–A interaction and conjugation length, which possessed the functions of NIR-II fluorescence imaging, efficient type-I PDT and PTT properties [104]. As displayed in Figure 8, the introduction of the diphenylamine unit on the conjugated small molecule (CTBT) backbone yielded a distorted conformation, resulting in a DCTBT molecule with a much better AIE property. DCTBT NPs showed maximal absorption at 704 nm and emission at 995 nm with a high fluorescence QY of 4.37%. By using different kinds of ROS indicators, it was demonstrated that DCTBT NPs could mainly produce O_2_^•−^ under 808 nm laser exposure, suggesting the type-I PDT process. The photothermal effect was further evaluated with 808 nm laser irradiation, and a high PCE of 59.6% was measured. DCTBT was assembled into liposomes by doping the EGFR-targeting peptide-modified amphiphilic polymer DSPE-PEG_2000_-GE11 as the encapsulation matrix to promote effective aggregation and visualization of lip-DCTBT NPs at tumor sites. In vivo NIR-II fluorescence imaging of subcutaneous PANC-1 tumor-bearing mice helped to precisely delineate the tumor site and further phototherapy exhibited significant tumor growth inhibition. More interestingly, DCTBT NPs were also able to suppress the growth of orthotopic pancreatic tumors under synergistic NIR-II fluorescence-guided type-I PDT and PTT treatments.

## 5. Photodynamic NIR-II AIEgens for Both Tumor and Bacteria Inhibition

Recent studies have shown a close relationship between bacteria and human cancer cells, including the promotion of cancer cell development and metastasis [105,106]. Therefore, a highly efficient system that could simultaneously kill bacteria and cancer cells would benefit tumor treatment. To achieve this, Sun and Kim et al., reported a NIR-II phototheranostic agent with an AIE property, which consisted of two parts: a D-A-D scaffold with NIR-II fluorescence/PA imaging signals and associated PDT and PTT properties, and a widely used AIE building block (TPE) with excellent emission efficiency (Figure 9) [107]. The ZSY-TPE compound exhibited an AIE feature with maximal absorption/emission at 730/1020 nm, and a large Stokes shift of about 290 nm. ZSY-TPE NPs possessed a good photothermal effect under the irradiation of an 808 nm laser with a PCE of 28%, enabling a strong PA signal. More interestingly, the AIE NPs could also generate ROS upon 808 nm laser irradiation, which had ^1^O_2_ generation quantum yield of 13.8% with ICG (12%) as the standard. For the excellent NIR-II fluorescence properties, as well as the good photodynamic, photothermal, and PA properties, the AIEgen was explored for imaging-guided PDT and PTT of tumors and pathogens. After 24 h post-injection of the AIE NPs into 4T1 tumor-bearing mice, the tumor region was greatly illuminated by NIR-II fluorescence and PA imaging. In contrast to the single modal imaging, this dual NIR-II/ PA imaging helped to provide complementary information about the tumor and precisely guide the subsequent phototherapy. Under the irradiation of an 808 nm laser, the tumor growth was significantly inhibited thanks to the combination of PDT and PTT. Moreover, the AIE NPs could also be used for the NIR-II fluorescence imaging of *Staphylococcus aureus*-infected mice, and the following imaging-guided PDT/PTT was performed to inhibit bacterial infections. Considering the close relationship between bacteria and cancers, this kind of multifunctional phototheranostic agent may represent an efficient strategy for precise diagnostics and therapeutics of bacteria-infected tumors.

## 6. Hybrid Self-Assembly System for Phototheranostics

Recently, Wang and Tang et al., reported a prismatic metal cage C-DTTP with bright NIR-II fluorescence emission through the assembly of an AIE-active four-armed ligand with a 90° Pt acceptor Pt(PEt_3_)_2_(OTf)_2_ [108]. As displayed in Figure 10, a D-A-D type molecule (DTTP) was rationally designed with maximal absorption/emission at 675/993 nm. The four pyridine substitutes endowed DTTP with the function of a four-armed ligand, which could form the metal-cage C-DTTP via supramolecular coordination. C-DTTP exhibited remarkable AIE characteristics with a maximum emission wavelength of 1005 nm, which was the longest fluorescence emission wavelength compared with the previously reported supramolecular coordination complexes (SCCs). The fluorescence QY of the mPEG-PLGA-encapsulated C-DTTP NPs (CNPs) was 1.61%, and it also showed a PCE of 39.3% under the irradiation of the 808 nm laser. Moreover, CNPs exhibited a much higher ROS generation efficiency than LNPs (the ligand DTTP-constituted NPs). The in vivo imaging-guided PDT/PTT treatment was performed on MDA-MB-231 tumor-bearing BALB/c nude mice. After intravenous injection of CNPs, the NIR-II fluorescence signal in the tumor area gradually increased with time, and reached the highest intensity at about 12 h. Subsequently, the anti-tumor effect of CNPs was studied. The treatment of CNPs with 808 nm laser irradiation achieved complete tumor clearance on the 14th day, which suggested excellent anti-tumor effect. This study provides an example of creating NIR-II emitting SCCs with unified diagnostic and therapeutic properties, which represents a new way for promoting the biomedical applications of SCCs.

## 7. Summary and Perspective

As NIR-II fluorophores possess narrow electronic bandgaps, the low energy levels are usually not enough for generating ROS under light irradiation. With the endeavor of many researchers, NIR-II AIEgens with a PDT property are emerging in the last two years. The low bandgaps of NIR-II emitters result in dominated nonradiative thermal deactivation of the excited state energy in most cases. Therefore, NIR-II chromophores are naturally born with photothermal and PA properties, which enables multifunctional phototheranostics. In this minireview, we summarize the recent advances of photodynamic NIR-II AIEgens and their applications (Table 1). The molecular design strategies for tuning the electronic bandgaps and photophysical properties are discussed. We also highlight the biomedical applications such as image-guided therapy of both subcutaneous and orthotopic tumors, and multifunctional theranostics in combination with other treatment methods, including chemotherapy and immunotherapy; and the precise treatment of tumor and bacterial infection. This kind of agents turn out to be powerful for high-resolution diagnosis and precise disease therapeutics. Some aspects can be considered for their future development. First, the study of photodynamic NIR-II AIEgens is still in the infancy; more systematic investigations are needed to provide a comprehensive understanding about these kinds of molecules. Second, although NIR-II luminogens possess multifunctional properties in one molecule, new strategies that could tune and optimize each imaging/therapy modality as needed are of great significance to boost the theranostic outcome. Third, as the current applications mainly focus on tumors, more research in other diseases should be explored to extend the applications. Last, since this is a new kind of material, the long-term biocompatibility should be carefully evaluated to push forward the clinical transformation. This review aims to provide guidance for the PDT agents with long-wavelength emission to improve the diagnostic precision and treatment outcome. We hope it will provide a comprehensive understanding about the chemical structure–photophysical property–biomedical application relationship of NIR-II luminogens.

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
