# Peer review of "NIR-II AIEgens with Photodynamic Effect for Advanced Theranostics"

_molecules, 2022, doi:10.3390/molecules27196649_

Round 1

Reviewer 1 Report

Recently, many reviews have been published focusing on AIE dyes (according to theweb of science about 120 reviews with the word AIGens in the last 3 years), i.e. 2022 in Coordination Chemistry Reviews, Nanotechnology 2021, 2021 Aggregate, 2021 Materials Chemistry Frontiers, 2018 Theranostic. This means that research work on AIGEns is constantly developing in all directions, from synthesis to theransotic applications. However, the review by Qi et al. focuses on the latest work on AIEgens useful in photodynamic therapy. The authors have systematically reported the selected papers and illustrate them with many figures, reproduced with permission. As consequence, the presented review looks more like a report than a comprehensive and critical review. There is no comparison between the systems presented, for example some summary table could be useful. As a reader, I don't know which system is better for diagnostics and which for treatment, or which for both applications. Many figures contain the same list of AIGens’ advantages or a similar JabÅ‚oÅ„ski diagram or an image of a mouse with a syringe and I think the repeatable elements should be eliminate.

Author Response

Response: We sincerely thank the reviewer for your careful reading and positive feedback to our manuscript. According to the reviewer’s suggestion, we have provided a summary table to compare the properties of the molecules in the revised manuscript (new Table 1). We have also modified some figures in the revised manuscript (Figure 8).

Reviewer 2 Report

In the review manuscript "NIR-II AIEgens with Photodynamic Effect for Advanced Theranostics", the authors summarized the recent advances of photodynamic NIR-II aggregation-induced emission lunimogens (AIEgens) for biomedical applications. They discussed the molecular design strategies for tuning the electronic bandgaps and photophysical energy transformation processes. They also highlighted the biomedical applications such as image-guided therapy of both subcutaneous and orthotopic tumors, multifunctional theranostics in combination with other treatment methods including chemotherapy and immunotherapy, and precise treatment of both tumor and bacterial infection. The topic is very important for scientists interested in long-wavelength optical imaging and therapeutics. This manuscript is well prepared and the overall quality is high. So I think it can be accepted for publication in Molecules after minor revision. More comments are shown below:

1. There is no definition for some scale bars, please provide the definition in order to be easy for the readers to understand.

2. The authors should pay attention to the abbreviations of some compounds. For example, both DCFH and DCFH-DA are used in the manuscript, please double check them.

3. It seems there is some confusion with the description and arrangement of Figure 7 and Figure 8, please correct them.

4. In Figure 9, there are only a-f six figures, but there are seven legends (a-g) in the figure caption. Please revise them.

5. Please pay attention to the reference format. For example, there are “et al.” for the author lists of some references, please check them according to the journal Instructions for Authors.

Author Response

In the review manuscript "NIR-II AIEgens with Photodynamic Effect for Advanced Theranostics", the authors summarized the recent advances of photodynamic NIR-II aggregation-induced emission lunimogens (AIEgens) for biomedical applications. They discussed the molecular design strategies for tuning the electronic bandgaps and photophysical energy transformation processes. They also highlighted the biomedical applications such as image-guided therapy of both subcutaneous and orthotopic tumors, multifunctional theranostics in combination with other treatment methods including chemotherapy and immunotherapy, and precise treatment of both tumor and bacterial infection. The topic is very important for scientists interested in long-wavelength optical imaging and therapeutics. This manuscript is well prepared and the overall quality is high. So I think it can be accepted for publication in Molecules after minor revision. More comments are shown below:

  1. There is no definition for some scale bars, please provide the definition in order to be easy for the readers to understand.

Response: We sincerely thank the reviewer for your careful reading and positive feedback to our manuscript. We have provided the definition of scale bars in the revised manuscript.

  1. The authors should pay attention to the abbreviations of some compounds. For example, both DCFH and DCFH-DA are used in the manuscript, please double check them.

Response: We thank for the reviewer’s suggestion. We have carefully checked and revised the abbreviations of compounds in the revised manuscript.

  1. It seems there is some confusion with the description and arrangement of Figure 7 and Figure 8, please correct them.

Response: We have corrected the figures in the revised manuscript.

  1. In Figure 9, there are only a-f six figures, but there are seven legends (a-g) in the figure caption. Please revise them.

Response: We have corrected the figure caption in the revised manuscript.

  1. Please pay attention to the reference format. For example, there are “et al.” for the author lists of some references, please check them according to the journal Instructions for Authors.

Response: According to the reviewer’s suggestion, we have checked and revised the references according to the journal Instructions for Authors in the revised manuscript.

Reviewer 3 Report

In the present revision by Yin et al. the authors present relevant examples of NIR-II AIE luminogens and how distinct molecular design strategies endow these structures with relevant photoluminescence properties including PDT ability. This is an important contribution to advances in theranostics by which light can be used both as a means for diagnosis and treatment enabling improved imaging precision and treatment efficacy.

The text is well written and to an adequate extent, the set of figures selected provides the necessary information and it is well supported by an adequate set of references.

Nonetheless, in my opinion, some points have to be addressed before publication:

-          In the abstract, line 16, the word “window” should be placed before the brackets

-          In the abstract, line 21, the word “lunimogens” has to be corrected to “luminogens”;

-          In the Introduction, line 45, the expression “For examples” should be replaced by “For example” or “Namely”.

-          In the Introduction, line 76, in “benefitting for…” withdraw “for”.

-          In the Introduction, line 95, replace “small energy …” with “small energy…”.

-          In figure 1 legend, the meaning of “DDPD” and “TPE” should be provided, since it is the first time it appears in the text.

-          The main body of the manuscript is in section 2. As a means of clarity and ease of accessibility to its contents, I believe that it would benefit from the existence of subsections. As a mere example, it could be extra functionality/application provided by the system in combination with PDT such as chemotherapy (Fig. 5); immunotherapy (Fig. 6); PAI (Fig. 2, 3, and 4); simultaneous killing of bacteria and cancer cells (Fig. 9); orthotopic tumors treatment (Fig. 8); hybrid self-assembly (Fig. 10).

-          In section 2, lines 125-128, the definition provided for Type I PDT should be rephrased. It is not highlighted that in this case, it is not a direct reaction with molecular oxygen.

-          In section 2, line 146, as well as in other cases ahead, the reference to several authors followed by et al. should be corrected to the first author's surname followed by et al.

-          The text and figures from lines 282-304 refer to figure 8. But those concerning figure 7 appear only in lines 305 – 322.

-          The caption of figure 9 does not seem to correspond to the data presented.

-          Line 327, “which was consisted of” remove “was”.

-          The references section should be uniformized, e.g.: words in titles should appear either capitalized like in ref 86 or just the first word like in ref 78.

Author Response

In the present revision by Yin et al. the authors present relevant examples of NIR-II AIE luminogens and how distinct molecular design strategies endow these structures with relevant photoluminescence properties including PDT ability. This is an important contribution to advances in theranostics by which light can be used both as a means for diagnosis and treatment enabling improved imaging precision and treatment efficacy.

The text is well written and to an adequate extent, the set of figures selected provides the necessary information and it is well supported by an adequate set of references.

Nonetheless, in my opinion, some points have to be addressed before publication:

- In the abstract, line 16, the word “window” should be placed before the brackets

- In the abstract, line 21, the word “lunimogens” has to be corrected to “luminogens”;

- In the Introduction, line 45, the expression “For examples” should be replaced by “For example” or “Namely”.

- In the Introduction, line 76, in “benefitting for…” withdraw “for”.

- In the Introduction, line 95, replace “small energy …” with “small energy…”.

Response: We sincerely thank the reviewer for your careful reading and positive feedback to our manuscript. We have corrected these problems according to the reviewer’s great suggestion.

- In figure 1 legend, the meaning of “DDPD” and “TPE” should be provided, since it is the first time it appears in the text.

Response: Yes, we have provided the full names of these abbreviations in the revised manuscript.

- The main body of the manuscript is in section 2. As a means of clarity and ease of accessibility to its contents, I believe that it would benefit from the existence of subsections. As a mere example, it could be extra functionality/application provided by the system in combination with PDT such as chemotherapy (Fig. 5); immunotherapy (Fig. 6); PAI (Fig. 2, 3, and 4); simultaneous killing of bacteria and cancer cells (Fig. 9); orthotopic tumors treatment (Fig. 8); hybrid self-assembly (Fig. 10).

Response: According to the reviewer’s suggestion, we have divided the main body into several sections according to the function in the revised manuscript.

- In section 2, lines 125-128, the definition provided for Type I PDT should be rephrased. It is not highlighted that in this case, it is not a direct reaction with molecular oxygen.

Response: We thank for the reviewer’s suggestion, and we have corrected the definition of type I PDT in the revised manuscript (line 129-131).

- In section 2, line 146, as well as in other cases ahead, the reference to several authors followed by et al. should be corrected to the first author's surname followed by et al.

Response: According to the reviewer’s suggestion, we have corrected them in the revised manuscript.

- The text and figures from lines 282-304 refer to figure 8. But those concerning figure 7 appear only in lines 305 – 322.

Response: We are sorry for the mistake, and we have corrected them in the revised manuscript.

- The caption of figure 9 does not seem to correspond to the data presented.

Response: We have corrected the figure caption of Figure 9 in the revised manuscript.

- Line 327, “which was consisted of” remove “was”.

Response: Corrected.

- The references section should be uniformized, e.g.: words in titles should appear either capitalized like in ref 86 or just the first word like in ref 78.

Response: We thank for the reviewer’s suggestion. The words in titles of the references have been capitalized in the revised manuscript.